# On the Value of Out-of-Distribution Testing: An Example of Goodhart's Law

**Damien Teney**[1]    **Kushal Kafle**[2]    **Robik Shrestha**[3]
**Ehsan Abbasnejad**[1]    **Christopher Kanan**[3]    **Anton van den Hengel**[1]

[1]Australian Institute for Machine Learning, University of Adelaide, Australia
[2]Adobe Research
[3]Rochester Institute of Technology

{damien.teney,ehsan.abbasnejad,anton.vandenhengel}@adelaide.edu.au
kkafle@adobe.com   {rss9369,kanan}@rit.edu

## Abstract

Out-of-distribution (OOD) testing is increasingly popular for evaluating a machine learning system's ability to generalize beyond the biases of a training set. OOD benchmarks are designed to present a different joint distribution of data and labels between training and test time. VQA-CP has become the standard OOD benchmark for visual question answering, but we discovered three troubling practices in its current use. First, most published methods rely on explicit knowledge of the construction of the OOD splits. They often rely on "inverting" the distribution of labels, e.g. answering mostly "yes" when the common training answer is "no". Second, the OOD test set is used for model selection. Third, a model's in-domain performance is assessed after retraining it on in-domain splits (VQA v2) that exhibit a more balanced distribution of labels. These three practices defeat the objective of evaluating generalization, and put into question the value of methods specifically designed for this dataset. We show that embarrassingly-simple methods, including one that generates answers at random, surpass the state of the art on some question types. We provide short- and long-term solutions to avoid these pitfalls and realize the benefits of OOD evaluation.

## 1    Introduction

> Goodhart's law:   *When a measure becomes a target, it ceases to be a good measure.*

The practical value of a machine learning (ML) system is strongly related to its capacity to generalize, *i.e.* to produce relevant outputs for data beyond its training set. The common paradigm in learning theory [42] assumes that the training and test data are drawn as independent and identically distributed (IID) samples. Therefore, most datasets are built following this IID assumption, such that data points are assigned randomly to training or test splits. For many tasks however, this fails to assess whether an ML system adequately generalizes, or whether it has simply captured the idiosyncrasies of a dataset, including spurious correlations that manifest in both the training and test sets [28].

Out-of-distribution (OOD) testing is an increasingly popular method for evaluating generalization [3, 4, 7, 23, 25, 29]. In this paper, we use "OOD" to refer to splits designed such that the joint distribution of inputs and labels differs between the training and testing sets. The differences in this distribution concern features of the data that are irrelevant to the task of interest, *e.g.* the background in an image recognition task. Such irrelevant factors can be spuriously correlated with the correct labels. They form "dataset biases" and other statistical patterns that a robust model should not rely on.

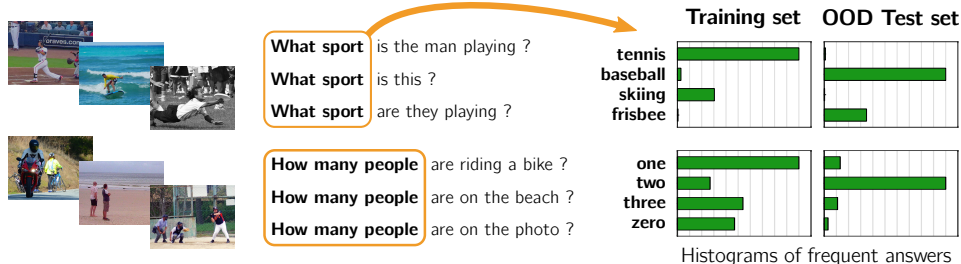

Figure 1: In the VQA-CP dataset, the distribution of answers given a question prefix differs between training and testing. We show that many existing methods exploit the fact that **the training and test distributions are approximately inverse of each other**. This is made possible through the bad practice of using the OOD test set for model selection. Moreover, the issue is completely hidden from the in-domain evaluation typically performed after retraining on the VQA v2 dataset, because its distribution of answers is more uniform.

This paper takes a close look at VQA-CP [3], an OOD benchmark for visual question answering (VQA). In VQA, a model is provided with an image and a related question and must produce a relevant answer. VQA is usually approached as a classification problem over a large set (1000+) of candidate answers, and usually trained with a large dataset of questions and correct answers [39, 44]. These datasets are produced by human annotators and contain strong biases. For example, most questions of the form *Is there a ... in the image ?* are correctly answered with *yes* [6]. The VQA-CP dataset was designed to evaluate models in an OOD setting. It was built by re-splitting the VQA v2 dataset [20] such that the joint distribution of answers and question prefixes differs between training and testing (see Fig. 1). Considerable effort has been put into methods specifically designed to improve performance on VQA-CP [3, 10, 12, 22, 32, 36, 41]. Unfortunately, we discovered multiple flaws in the experimental setup and design of many of these methods.

This paper exposes three critical issues. **First**, almost all published methods evaluated on VQA-CP are designed for high performance specifically on its OOD test set. They rely on the known construction procedure of the OOD splits (see Fig. 1). **Second**, since the dataset has no official validation set, almost all methods use the OOD test set for model selection. **Third**, the common practice is to verify that a model performs well on in-domain data, but this is performed after retraining on standard splits (VQA v2), which exhibit a different label distribution.

This paper discusses how these issues defeat the purpose of an OOD evaluation. The situation is a striking example of Goodhart's law: the metrics of the benchmark are gamed in a way that defeats its original, well-intended purpose. Instead of making advances towards generalization in vision-and-language, the performance on VQA-CP has been treated as a standalone objective. This puts the value of many published methods into question. To demonstrate this point, we present several embarrassingly-simple baselines (including one that draws answers at random) that surpass the state of the art on some or all question types on VQA-CP.

In summary, the contributions of this paper are as follows.
1. We highlight three major flaws in the experimental setup and in the design of methods for VQA-CP, and we discuss how they amount to subtly cheating the OOD evaluation.
2. To demonstrate the point concretely, we describe and evaluate embarrassingly-simple methods that surpass published results on VQA-CP on some or all question types.
3. We provide guidelines for the continued use of VQA-CP to best capture the benefits of OOD evaluation. We also point at promising directions for the design of future benchmarks.

## 2   Background

### 2.1   Visual question answering

VQA involves answering a text question $q$ about an image $v$ with an answer $a^{\mathrm{pred}}$. It is typically treated as a classification task over a large set of candidate answers [39, 44]. Formally, a VQA model is a function $f(q, v) = a^{\mathrm{pred}}$ where the question $q$ is a vector of tokens from a predefined vocabulary, the visual features $v$ are a set of vectors representing visual features of object detection [5], and the output $a^{\mathrm{pred}} \in \mathbb{R}^K$ is a vector of predicted scores over a large set of $K$ predefined answers. This function $f(\cdot)$ is typically implemented as a neural network and trained with supervision on a training

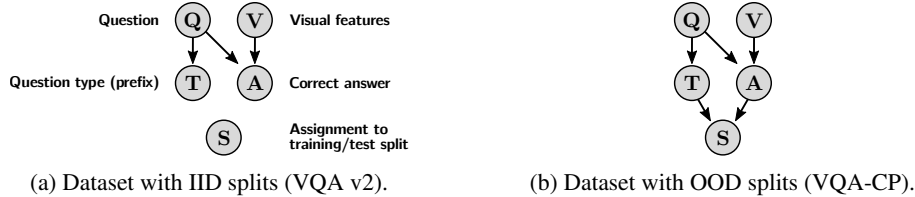

(a) Dataset with IID splits (VQA v2).      (b) Dataset with OOD splits (VQA-CP).

Figure 2: Statistical dependencies in VQA datasets. When creating VQA-CP (b), the question type ($T$) and answers ($A$) serve to assign data points to the training or test split ($S$). A typical VQA model is trained to approximate $P(A \,|\, Q, V)$. Many methods for VQA-CP exploit dataset-specific priors, additionally conditioning on $T$ and $S$ and learning $P(A \,|\, Q, V, T, S)$.

set $\mathcal{T}$ of triplets made of questions, images, and their correct answers: $\mathcal{T} = \{(\boldsymbol{q}_i, \boldsymbol{v}_i, \boldsymbol{a}_i^{\text{gt}})\}_{i=1}^N$. The ground truth answers $\boldsymbol{a}^{\text{gt}} \in (0, 1)^K$ are vectors of scores over the same set of candidate answers as $\boldsymbol{a}^{\text{pred}}$. We use upper-case letters to refer to the random variables representing distributions over the whole dataset, such that $\boldsymbol{q}_i \sim Q$, $\boldsymbol{v}_i \sim V$ and $\boldsymbol{a}_i^{\text{gt}} \sim A$.

## 2.2 Dataset biases

The most popular dataset, VQA v2 [20], is built from human annotators. Therefore, it inevitably displays strong biases. This includes **static biases** *i.e.* non-uniform distributions over individual modalities (images, questions, or answers). For instance, questions such as *What is the person doing* are extremely frequent. The image distribution is also biased. They come from COCO/Flickr [30] and depict a surprising number of people surfing and playing Wii, for example. The images also focus on the 80 object classes annotated in COCO. The dataset also features **conditional biases**, *i.e.* non-uniform conditional distributions. The one most discussed in the literature, known as the language bias, refers to the distribution of answers given the questions. It makes answers easy to guess without considering the image [3, 21, 28], a form of "shortcut learning" [18].

## 2.3 The VQA-CP dataset

The VQA-CP dataset (for Changing Priors) was designed to evaluate VQA models in a setting where they cannot rely on language biases. The benchmark penalizes models for predictions that agree with the language biases present in the training data. The dataset was **built by reorganizing the training/test splits of VQA v2** as follows. The questions are assigned to one of 65 question types according to their prefix (first few words). The prefixes were defined in [20]. All question/image/answer triplets are then clustered according to the combination of prefix and answer. The clusters are randomly assigned to the training/test splits, while ensuring that most words appearing in test questions also appear in some of the training questions (see [3] for details).

To formally understand the issues with VQA-CP, it is useful to identify the **statistical dependencies within the dataset**. In Fig. 2a, we use Bayesian networks to represent the dependencies between the distributions of questions, visual features, and answers, *i.e.* the random variables $Q$, $V$, and $A$, respectively. The random variables represent distributions over the union of all splits of a dataset. The examples in the dataset are samples for an underlying distribution $P(A, Q, V)$.

– In VQA v2, the samples are assigned randomly to the training and test splits. We formalize this as an additional random variable $S \in \{\text{train}, \text{test}\}$. Since $S$ is independent, we have $P(A \,|\, Q, V, S\text{=train}) \approx P(A \,|\, Q, V, T\text{=test})$. This means there is no significant distribution shift between the training and test sets.

– In VQA-CP, in contrast, the assignment of a sample to the training or test split depends on its question type $t \sim T \in \{1, 2, ..., 65\}$ and answer $\boldsymbol{a}^{\text{gt}} \sim A$. Now $S$ is dependent on $T$ and $A$ (Fig. 2b). This fulfills the aim of creating OOD splits, which is to make the joint distribution of questions and answers differ between training and test time *i.e.* $P(Q, A) \neq P(Q, A \,|\, T)$.

When learning a VQA model, one normally seeks to approximate $P(A \,|\, Q, V)$. The first issue discussed below (Section 3.1) is that models for VQA-CP are often conditioned on $T$ and/or $S$, *i.e.* they learn $P(A \,|\, Q, V, T, S)$. This fits the data-generating process of VQA-CP but it exploits dependencies that are idiosyncratic to VQA-CP.

## 3 Existing methods for VQA-CP and their issues

The VQA-CP dataset has sparked much research to address the excessive reliance on the language bias of existing models. Several proposed methods rely on compensating for question-answer distribution patterns [10, 12, 14, 36, 22]. This is typically achieved with a regularizer based off an auxiliary model that is trained to predict answers from the question alone, without access to image features [10, 12, 22, 36]. Other works [37, 43] have used additional supervision from human attention maps to encourage a model to use relevant image regions. Jing *et al.* [26] proposed to use the question to pre-select a shortlist of potential answers and relevant image regions, and to predict the final answer with a separate module having no direct access to the question. Other recent works focused on synthesizing counterfactual examples to improve robustness in VQA-CP [1, 11, 13].

Table 1: Selection of methods and their main issues.

| | Issue 1 | | Issue 2 | Issue 3 |
|---|---|---|---|---|
| | Rely on dataset construction | Use question type/prefix | Use test set for model selection | Retrain for in-domain evaluation |
| *Question-based regularization* | | | | |
| Ramakrishnan *et al.* [36] | 😠 | 🙂 | 😠 | 😠 |
| Grand and Belinkov [22] | 😠 | 🙂 | 🙂 | 😠 |
| RUBi [10] | 😠 | 🙂 | 😠 | 😠 |
| Learned-Mixin [12] | 😠 | 😠 | 😠 | 😠 |
| *With additional annotations to improve visual grounding* | | | | |
| HINT [37] | 🙂 | 🙂 | 😠 | 😠 |
| SCR [43] | 🙂 | 🙂 | 😠 | 😠 |
| *Other methods* | | | | |
| GVQA [3] | 🙂 | 😐 | 😠 | 😠 |
| Decomposed linguistic repr. [26] | 😠 | 😠 | 😠 | 😠 |
| Actively Seeking [41] | 🙂 | 🙂 | 🙂 | 🙂 |
| Unshuffling [14] | 😠 | 😐 | 🙂 | 🙂 |
| Gradient supervision [13] | 🙂 | 🙂 | 🙂 | 🙂 |

The performance of methods proposed for VQA-CP has steadily increased over time. However, most of these methods were specifically designed for VQA-CP (*e.g.* [10, 22, 36, 26]). We argue that targeting the improvement of accuracy on VQA-CP as a sole objective is misguided. Indeed, we have identified three common practices that allow one to obtain a high accuracy while sidestepping the original intent of the benchmark. A summary of existing methods and their issues is given in Table 1.

### 3.1 Issue 1: Relying on the known construction of the OOD splits

Most methods for VQA-CP **exploit the knowledge that the answers are distributed differently in the test set according to the questions' prefix** (Section 2.3), either by design or inadvertently. This includes methods that simply aim to reduce the general dependence of the predicted answer on the question [10, 26, 36] or, more alarmingly, methods that directly use the question prefix information; even to the point of using the ground truth annotations that was originally used to create the dataset[26]. While this is effective for maximizing the accuracy on the test set, the resulting model is unlikely to generalize beyond the particular setup of VQA-CP. Many of these methods would either fail or require considerable rework if the dataset had bias stemming from other sources (sentiment, object-attribute, images, *etc.*).

Teney *et al.* [14] acknowledge the issue. The "unshuffling" technique can be applied on multiple sets of feature, but, unsurprisingly, unshuffling using the ground truth question prefix maximizes the accuracy on the VQA-CP test set. While the same technique could be used to reduce reliance on other types of biases, it would not be visible on VQA-CP.

A related, but more subtle issue than using the prefixes, is to exploit the fact that **the distributions of answers given a prefix are approximately inverse of each other** between training and testing (see Fig. 3, and 6 in the supplementary material). This systematic relation is highly artificial but and exploiting it is very effective on this particular dataset (see Section 5). More troublesome, a method may rely on this heuristic only implicitly, as a consequence of issue 2.

### 3.2 Issue 2: Using the test set for model selection

Many methods use the OOD test set of VQA-CP in place of a validation set since the dataset only defines official training and test sets. This goes against widely accepted "best practice" in ML to reserve the test set for the final evaluation of a model so as to avoid adaptive overfitting [9, 15]. A validation set is a practical necessity for debugging, tuning hyperparameters, early stopping, *etc.*. OOD benchmarks bring the additional challenge that an ideal validation set cannot be trivially held

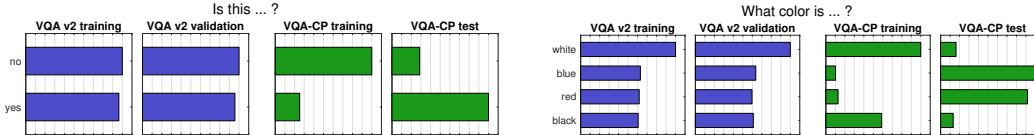

Figure 3: Histograms of the most frequent answers for examples of question prefixes in VQA v2 and VQA-CP. The training/test distributions in VQA-CP are approximately inverses of one another. This artefact is often exploited to obtain artificially-strong performance, explicitly (**issue 1**) or implicitly (**issue 2**). Unfortunately, in-domain evaluation on VQA v2 does not reveal this behaviour (**issue 3**) because it displays a more uniform distribution of answers (additional examples in supp. mat.).

out from the training data, because it would not reflect OOD conditions. The absence of an OOD validation set has been used as justification to use the test set for model selection by authors who do recognize it to be problematic [22]. Others [22, 14, 41] build a validation set held out from the training data. Even though it cannot predict OOD performance, the authors reported it to be suitable for hyperparameter tuning [14] and early stopping [22].

In Section 5, we show that hyperparameter tuning and early stopping have a massive influence on the accuracy on *yes/no* and *number* (Nb) questions. Using the test data for model selection is therefore a subtle form of overfitting and it leads to an artificially inflated performance on the test set. Since an OOD benchmark fundamentally aims at **simulating a situation where the distribution of the test data is unknown**, this practice defeats the purpose of the benchmark. In comparison to standard IID splits, the risk of adaptive overfitting is magnified with OOD splits because the optimal hyperparameters best suited to the test set are likely to greatly differ from those that are best for the training data.

### 3.3 Issue 3: Evaluating in-domain performance after retraining

OOD Performance is essentially a proxy measure of generalization, and it is therefore important that a model remains *simultaneously* effective on in-domain data. However, almost all works evaluated on VQA-CP monitor in-domain performance after retraining the model on VQA v2 which effectively means evaluating two different copies of the model, each optimized on a different training set. The intention is to make the in-domain performance serve the separate purpose of being comparable with existing models evaluated on VQA v2. The subtle issue is that **a method can behave differently depending on the distribution of labels in the training data**.

This is precisely the case between VQA v2 and VQA-CP. VQA v2 was specifically designed such each question is paired with two images leading to different answers. Antagonist answers like *yes* and *no*) are thus similarly likely. VQA-CP, by construction, breaks this property (see Fig. 3).

As detailed in issue 1, many methods invert the distribution of answers between training and testing (implicitly or explicitly). This means they rely on the non-uniform distribution of answers in VQA-CP. When retrained on VQA v2, the training distribution is more balanced and the effect of these methods is correspondingly scaled down. As shown in Section 5, the performance of methods trained on VQA-CP significantly drop on in-domain data which indicates limited benefits in overall generalization). This effect is however not visible when retraining on VQA v2.

## 4 Proposed methods

We describe simple, strong baselines on VQA-CP. They are of no real-world utility by construction. They serve to evaluate the extent to which high performance can be obtained by exploiting the issues described above.

- **Random predictions.** This method samples answers at random according to the distribution of answers **observed per individual question type** in the training set. A question $q$ is associated to a type $t \in \{0, 1, ..., 65\}$ by trivially matching its first few words with the set of 65 prefixes defined in [20]. We use a function $t = \text{matchPrefix}(q)$ that performs longest-string matching with these known prefixes. The method is "trained" by accumulating an empirical estimate of $P(A \mid T)$ over the training set, *i.e.* a 2D histogram $\boldsymbol{H} \in \mathbb{N}^{65 \times K}$. At test time, for a question $q$ of prefix $p$, the method samples an index $k$ of a candidate answer from $P(A \mid T=t)$. The method outputs a one-hot vector of scores $\boldsymbol{a}^{\text{pred}}$ such that $\boldsymbol{a}^{\text{pred}}[k] = 1$ and 0 elsewhere, where $[k]$ denotes the $k^{\text{th}}$ element.

- **Random predictions, inverted.** This variation exploits the knowledge that the distribution over the test data is approximately proportional to the inverse of the distribution over the training data, *i.e.* $P(A \mid P, S=\text{train}) \propto 1/ P(A \mid P, S=\text{test})$. After accumulating $\boldsymbol{H}$ as described above, we compute $\bar{\boldsymbol{H}} = 1/\boldsymbol{H}$. The empty bins in $\boldsymbol{H}$ stay empty in $\bar{\boldsymbol{H}}$ (combinations of answers and question types never observed together in the training data). At test time, we sample answers using $\bar{\boldsymbol{H}}$ instead of $\boldsymbol{H}$.

- **Learned baseline.** This method is the standard bottom-up-top-down attention (BUTD) model [40] trained with a binary cross-entropy loss $\mathcal{L}_{\text{BCE}}$. Our notation $f(\cdot)$ represents the network without a sigmoid or softmax activation and $\boldsymbol{a}^{\text{pred}}$ represents the logits $\boldsymbol{a}^{\text{pred}} = f(\boldsymbol{q}, \boldsymbol{v})$.

- **Learned, top answer masked.** This variation exploits the heuristic that the answer most strongly correlated with a given question or image is unlikely to be the same in the training and test sets. The method assigns, at test time, the lowest possible score to the answer of highest predicted score, such that $\boldsymbol{a}^{\text{pred}}[k] \leftarrow -\infty$ where $k=\arg\max(\boldsymbol{a}^{\text{pred}})$.

- **Learned, with random-image regularizer.** This is a simple version of adversarial regularization [10, 12, 32, 36] similar to the "random premise" model [8] proposed for natural language inference. The goal is to discourage the model from making predictions that agree with the language bias $P(A \mid Q)$ observed during training. We augment the training data $\mathcal{T} = \{(\boldsymbol{q}_i, \boldsymbol{v}_i, \boldsymbol{a}_i^{\text{gt}})\}_{i=1}^N$ with a copy of the same questions paired with random visual features $\tilde{\boldsymbol{v}}_i$: $\mathcal{T}' = \{(\boldsymbol{q}_i, \tilde{\boldsymbol{v}}_i, \boldsymbol{a}_i^{\text{gt}})\}_{i=1}^N$. In practice, we get each $\tilde{\boldsymbol{v}}_i$ by randomly sampling from the visual features of other questions in the current mini-batch. We define an auxiliary loss to apply on $\mathcal{T}'$:

$$\mathcal{L}_{\text{aux}}(\boldsymbol{a}^{\text{pred}}, \boldsymbol{a}^{\text{gt}}) = \text{softmax}(\boldsymbol{a}^{\text{pred}})[k] \quad \text{where} \quad k = \arg\max(\boldsymbol{a}^{\text{gt}}) . \tag{1}$$

Minimizing $\mathcal{L}_{\text{aux}}$ encourages the score of the correct answer ($k$) to be low, since no supporting visual evidence is provided. The softmax makes this particular score depend on the scores of the other, incorrect answers. Therefore, the loss simultaneously encourages these other scores to be high. We minimize the main loss $\mathcal{L}_{\text{BCE}}$ over $\mathcal{T}$ and the auxiliary loss $\mathcal{L}_{\text{aux}}$ over $\mathcal{T}'$:

$$\min \left( \sum_{i=1}^N \mathcal{L}_{\text{BCE}} \left( f(\boldsymbol{q}_i, \boldsymbol{v}_i), \boldsymbol{a}_i^{\text{gt}} \right) \;+\; \lambda \sum_{i=1}^N \mathcal{L}_{\text{aux}} \left( f(\boldsymbol{q}_i, \tilde{\boldsymbol{v}}_i), \boldsymbol{a}_i^{\text{gt}} \right) \right) \tag{2}$$

with $\lambda$ a scalar hyperparameter. Each mini-batch contains an equal number of instances from $\mathcal{T}$ and $\mathcal{T}'$. This model is simpler to implement than existing versions of adversarial regularization since it uses the same architecture for the main and question-only predictions, and it does not requires gradient splitting or reversal.

## 5 Experiments

**Setup.** Each model is trained on VQA-CP and VQA v2. On VQA-CP, we hold out 8,000 instances from the training set (VQA-CP val.) to measure in-domain performance as proposed in [22, 14, 41]. Please refer to the supplementary material for additional details.

**Results.** Our **random predictions** give a relatively high accuracy on *yes/no/nb* on VQA-CP val. but low accuracy on VQA-CP test (see Table 2). This is expected: sampling from the training distribution is effective on the in-domain validation set but not on the OOD test set. The **random predictions, inverted** show the opposite. They give a remarkably high accuracy on VQA-CP test: >83% on *yes/no* and >49% on *nb*, both of which are superior to the state of the art on these questions. This accuracy however comes with a corresponding drop on the validation set. The trade-off between in-domain and OOD performance can only be assessed thanks to the validation set that we held out from the training data. The evaluation on VQA v2, which involves retraining the model and is the usual practice (rightmost columns in Table 2) completely hides this large drop in performance. Note also that the accuracy on *other* questions is near zero. Their set of plausible answers is so large that random sampling is unlikely to pick correct ones by chance. Random predictions have no practical utility, but these results demonstrate that **high accuracy on *yes/no/nb* questions can be obtained without any reasoning over text or images** by simply exploiting knowledge about the dataset that the existing methods also utilize.

We now look at the **learned** models. **Top answer masked** is remarkably effective on VQA-CP test: it improves from 43% (baseline) to 82% on *yes/no* and from 12% to 27% on *nb*. It is almost as effective on *yes/no* as the random predictions, while retaining some accuracy on the *other* questions. However, the accuracy on VQA-CP val. is extremely low. This trade-off between in-domain and OOD

Table 2: Accuracy of existing and proposed methods (%). Remarkably, the "**random predictions inverted**" surpass all other methods on *yes/no/number*. Highlighted cells are mentioned in the text. We recommend future comparisons to cite results trained on *other* questions alone (see supp. mat.).

| Training set → | VQA-CP Training | | | | | | | | VQA v2 Training | | | |
|---|---|---|---|---|---|---|---|---|---|---|---|---|
| Test set → | VQA-CP Val. (**in-domain**) | | | | VQA-CP Test (**OOD**) | | | | VQA v2 Val. (**in-domain**) | | | |
| | All | YesNo | Nb | Other | All | YesNo | Nb | Other | All | YesNo | Nb | Other |
| GVQA [3] | – | – | – | – | 31.30 | 57.99 | 13.68 | 22.14 | 48.24 | 72.03 | 31.17 | 34.65 |
| Ramakrishnan *et al.* [36] | – | – | – | – | 41.17 | 65.49 | 15.48 | 35.48 | 62.75 | 79.84 | 42.35 | 55.16 |
| Learned-Mixin [12] | – | – | – | – | 48.78 | 72.78 | 14.61 | 45.58 | **63.26** | **81.16** | **42.22** | **55.22** |
| Learned-Mixin+H [12] | – | – | – | – | 52.01 | 72.58 | 31.12 | 46.97 | 56.35 | 65.06 | 37.63 | 54.69 |
| RUBi [10] | – | – | – | – | 47.11 | 68.65 | 20.28 | 43.18 | 61.16 | – | – | – |
| Grand and Belinkov [22] | 56.90 | 69.23 | 42.50 | 49.36 | 42.33 | 59.74 | 14.78 | 40.76 | 51.92 | – | – | – |
| Actively seeking [41] | – | – | – | – | 46.00 | 58.24 | 29.49 | 44.33 | – | – | – | – |
| Unshuffling [14] | – | – | – | – | 42.39 | 47.72 | 14.43 | **47.24** | – | – | – | – |
| Gradient supervision [13] | 62.4 | 77.8 | 43.8 | 53.6 | 46.8 | 64.5 | 15.3 | 45.9 | 46.2 | 63.5 | 10.5 | 41.4 |
| Random predictions | 37.62 | 70.10 | 32.79 | 10.55 | 10.44 | 25.87 | 9.27 | 2.57 | 31.98 | 65.55 | 22.55 | 7.95 |
| **Random predictions, inverted** | 24.35 | 55.36 | 11.12 | 0.00 | 31.81 | 83.25 | 49.30 | 0.02 | 27.52 | 64.11 | 21.16 | 0.02 |
| Learned baseline (BUTD) | **64.73** | **79.45** | **49.59** | **55.66** | 38.82 | 42.98 | 12.18 | 43.95 | 59.93 | 77.66 | 36.85 | 52.41 |
| + Top answer masked | 30.90 | 44.12 | 25.00 | 20.85 | 40.61 | 82.44 | 27.63 | 22.26 | 31.84 | 48.22 | 25.78 | 20.62 |
| + RandImg λ=5 (best on CP test *other*) | 59.28 | 70.66 | 43.06 | 53.40 | 51.15 | 75.06 | 24.30 | 45.99 | 59.22 | 77.46 | 35.13 | 51.58 |
| + RandImg λ=12 (best on CP test *overall*) | 54.24 | 64.22 | 34.40 | 50.46 | **55.37** | 83.89 | 41.60 | 44.20 | 57.24 | 76.53 | 33.87 | 48.57 |

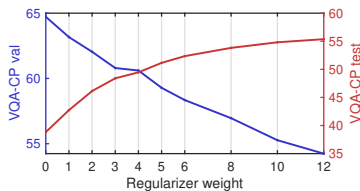

(a) Model trained on VQA-CP.

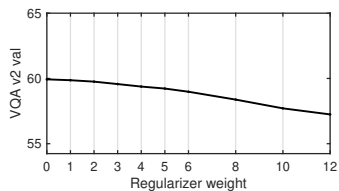

(b) Model retrained on VQA v2.

Figure 4: Accuracy of the random-image regularizer. A higher weight ($\lambda$) seemingly improves the accuracy on the OOD test set but the in-domain accuracy simultaneously drops. This can be assessed with the proposed held-out validation set (left), while the common practice of retraining on VQA v2 (right) makes the effect far less obvious. See the supp. material for a breakdown by answer type.

performance shows again that the performance of the model has not improved overall. The **random-image regularizer** is the closest of our methods to previously published works [10, 12, 22, 36]. The regularizer weight $\lambda$ allows tuning the trade-off between in-domain and OOD performance. We plot in Fig. 4 and Fig. 5 (in the supp. mat.) the accuracy as a function of $\lambda$. Three main observations can be drawn about this type of regularization. (1) It is most impactful on the *yes/no/nb* questions. With a large $\lambda$, the accuracy on the OOD test set approaches that of random predictions but does not surpass it. (2) It improves the accuracy on the *other* questions to a much smaller extent. There is a sweet spot for $\lambda$ that causes only a small drop in in-domain accuracy, suggesting an actual improvement in generalization. The sweet spot cannot be identified from the *overall* accuracy however, because it is dominated by the misleading effects on *yes/no/nb*. (3) The trade-off between in-domain and OOD performance is obvious with our held-out validation set, but it is near-impossible to assess through the usual practice of retraining on VQA v2.

Upon examination of **existing methods** (Table 2), we notice that a large fraction of claimed OOD improvements are attributable to *yes/no/nb* questions. Most methods are tuned for maximum *overall* accuracy, which puts emphasis on *yes/no/nb* rather than on the more difficult *other* questions. This means that **the results reported in most papers do not reflect the true potential of existing methods**[2] with the exception of those tuned in isolation on *other* questions [13, 14, 41]. For future reference, we report the performance of these methods, trained and evaluated on *other* questions alone (see Table 3 in the supplementary material).

# 6 Recommendations and discussion

**Recommendations for using VQA-CP.** We believe that VQA-CP remains one of several useful benchmarks to measure progress on VQA. We make recommendations for its continued use, aiming at capturing its original purpose of measuring generalization and resistance to language biases.

- The same model should be evaluated against in-domain and OOD data (same training, same hyperparameters). The current practice of retraining on VQA v2 is not acceptable to measure in-domain performance. We recommend **holding out a random subset** of 8k instances from the VQA-CP training data to evaluate in-domain performance to make it comparable to some existing works (see [41, 14] in Table 2).
- We recommend **focusing the analysis on *other* questions** exclusively. Questions with *yes/no/nb* answers are easier to game *e.g.* by implicit or explicit random guessing, while *other* questions are much less likely to be answered correctly with a naive or malfunctioning method. Training methods on *other* questions alone is acceptable, roughly halves the training time, and usually raises performance slightly on these questions (see supp. mat.).
- Care should be paid to the **analysis of the source of improvements** of any proposed method (as done retrospectively in [38] for HINT and SCR). For example, baselines should be tuned with as much care as the proposed method. The effect size of a regularizer should be shown to correlate with the weight of the regularizer. Quantitative improvements of small magnitude should be reported as averages across multiple runs with different random seeds.

**Recommendations for future datasets.** Evaluating generalization beyond the biases of a particular dataset is a challenging problem. We point at a few directions applicable beyond the context of VQA.

- Benchmarks with OOD test sets should cover **varying *levels* and *types* of distributional shifts**. A weakness of VQA-CP is to test for only one specific type of controlled correlation (question prefix/answer). An improvement would be to provide multiple training/test splits that probe the gamut from in-domain testing (IID splits) to extreme OOD, and across various confounders (*e.g.* correlations between visual concepts and language).
- A single aggregate metric may not be sufficient. In the case of multiple training/test splits, a model has to improve on a range of settings to be deemed valuable. Given the many ways in which current models can improve, the overall accuracy may not sufficient. **Multidimensional reports** (*e.g.* radar plots) may better convey the advantages of different methods in different settings.
- Another direction to evaluate generalization is to probe a model's behaviour densely near its decision boundaries. Datasets of **counterfactual examples** (a.k.a. contrastive) [2, 17, 13] contain alternate questions and/or images. They serve to verify that a model's decision changes in accordance to these interventions. This probes for a causal model of the predictions $P(\boldsymbol{a}^{\mathrm{pred}} \mid \mathrm{do}(Q, I))$ rather than the traditional correlation-based view $P(\boldsymbol{a}^{\mathrm{pred}} \mid Q, I)$ [34].

**Discussion.** This paper focused on three major issues that prevent realizing the full potential of OOD evaluation. Other issues exist, for example the possibility of subtle data leakage. We noted that the annotations of human attention used in [37, 43] allow accessing a distribution of labels during training closer to the OOD test set than to that of the original training set. The annotations are only available for a subset of the training questions that happens to be biased. The benefit that the methods using these annotations obtain from this bias has not been investigated, to our knowledge. Such subtle effects warrant greater care with OOD splits than with standard ones because a biased distribution is unlikely to provide an unfair advantage in the latter case.

A general solution to the subtle issues involved with OOD benchmarks is to raise the standards of scientific investigation [16], statistical reporting, and analysis [19, 24] as previously suggested in other areas of machine learning [31, 35]. The publication of empirical results often hinges on the scores obtained on top benchmarks, and yet a rigorous statistical analysis is seldom conducted [16, 24]. For example, few authors currently repeat their experiments with multiple random seeds, even when differences in performance across methods lie within the variance attributable to stochastic training. A lot of efforts in ML research are directed towards specific datasets and benchmarks, and it is therefore important that the experimental practices acceptable on these benchmarks incentivize valuable research. The onus falls not only on the dataset creators, but also on the authors of novel methods. Blindly optimizing a single metric is seldom a recipe for broadly applicable methods, and should rarely be the driving factor of a research project. The rising awareness of the need for more rigour in ML research is encouraging [16, 31, 33, 35]. We hope that this paper contributes to this trend of self-reflection and constructive criticism of common practices.

**Author contributions**

D.T. initiated the project, designed the proposed methods, ran the experiments, and wrote the initial draft of the paper. K.K. contributed to Sections 3 and 6, and helped editing the paper. R.S. contributed to Table 1, Sections 3 and 6, and helped editing the paper. E.A. contributed to Section 2 and helped editing the paper. C.K. contributed to editing the paper. A.H. contributed to editing the abstract and introduction.

**Broader impact**

By providing a better representation of the state of the art in visual question answering, and of current capabilities of AI systems, we believe this work will have a positive impact.

**Funding disclosure**

A.H. was employed partly by Amazon in Adelaide, Australia during the preparation of this manuscript. This company played no role in the sponsorship, design, data collection and analysis, decision to publish, or preparation of the manuscript.

C.K. and R.S. were supported in part by NSF award #1909696 to CK. The views and conclusions contained herein are those of the authors and should not be interpreted as representing the official policies or endorsements of any sponsor.

C.K. was employed at a commercial company, Paige, New York during the preparation of this manuscript. This company played no role in the sponsorship, design, data collection and analysis, decision to publish, or preparation of the manuscript.

## Footnotes

[1] There are non-intuitive implications to random predictions being better than a trained model. A model may perform best *before* training or in the early phases of optimization, rather than after convergence.

[2] A simple classifier can identify the type of answer (*yes/no*, *nb*, *other*) of >99% questions [27]. The accuracy of existing methods can thus be maximized by tuning them on *other* and using our strongest baseline for *yes/no/nb*.

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
