[Supplementary Material]

**Supplementary material to:**

**On the Value of Out-of-Distribution Testing: An Example of Goodhart's Law**

## A    Implementation details

### A.1    Implementation of the proposed methods

The **random predictions, inverted** model discard the answers observed in the training set below a fixed threshold, so as to avoid the inverse operation to assign extremely large probabilities to them. The answers retained after thresholding are: yes, no, 1, 2, 3, and 1,100 *other* answers. The **random-image regularizer** is applied on a BUTD model pretrained with the standard BCE loss for 25 epochs. Our methods based on the BUTD model use standard implementation and hyperparameters [40]. The BUTD model is optimized for maximum likelihood over the training set $\mathcal{T}$ with a standard binary cross-entropy (BCE) loss and a logistic function over the the the logits $\boldsymbol{a}^{\mathrm{pred}} = f(\boldsymbol{q}, \boldsymbol{v})$:

$$\mathcal{L}_{\mathrm{BCE}}\left(\boldsymbol{a}^{\mathrm{pred}}, \boldsymbol{a}^{\mathrm{gt}}\right) \;=\; -\,\boldsymbol{a}^{\mathrm{gt}} \log\left(\sigma(\boldsymbol{a}^{\mathrm{pred}})\right) \;-\; \left(1 - \boldsymbol{a}^{\mathrm{gt}}\right) \log\left(1 - \sigma(\boldsymbol{a}^{\mathrm{pred}})\right). \tag{3}$$

### A.2    Experimental setup

Our experiments follow the (flawed) common practice of training each model on VQA-CP (the *v2* version) then on VQA v2. We report performance at the epoch of highest accuracy on VQA-CP test or VQA v2 validation respectively. In addition, when training on VQA-CP, we hold out 8,000 instances from the training set that measure in-domain performance following [41, 14]. For existing methods, we report the results of highest overall accuracy on VQA-CP test reported by their respective authors.

## B    Additional results

See next page.

(a) Model trained on VQA-CP.

(b) Model trained on VQA v2.

Figure 5: Performance of the random-image regularizer as a function of the regularizer weight ($\lambda$) for (top to bottom): all questions, questions with *yes/no* answers, *number* answers, and *other* answers.

Table 3: Accuracy of methods trained and evaluated on VQA-CP 'Other' questions alone. Existing methods were implemented on top of different baseline models, so we report the accuracy of both the baseline and the proposed versions.

| Training set → | VQA-CP Training 'Other' | |
|---|---|---|
| Test set → | VQA-CP Val. 'Other' | VQA-CP Test 'Other' |
| Actively seeking [41]: baseline | 45.46 | 31.09 |
| Actively seeking [41]: as proposed | 46.79  (+2.33) | 34.25  (+3.16) |
| Gradient supervision [13]: baseline | 54.74 | 43.33 |
| Gradient supervision [13]: as proposed | **56.10**  (+1.36) | 44.70  (+1.40) |
| Unshuffling [14]: baseline | 54.74 | 43.33 |
| Unshuffling [14]: as proposed | 53.98  (-0.76) | **48.06**  (+4.73) |
| Random predictions | 10.39 | 2.63 |
| Random predictions, inverted | 0.03 | 0.06 |
| Learned baseline (BUTD) | **59.14** | 45.96 |
| Learned, top answer masked | 22.68 | 22.87 |
| Learned with random-image regularizer, $\lambda$=0 (=BUTD) | **59.14** | 45.96 |
| Learned with random-image regularizer, $\lambda$=1 | 58.19 | 46.60 |
| Learned with random-image regularizer, $\lambda$=2 | 57.52 | 47.85 |
| Learned with random-image regularizer, $\lambda$=3 | 57.60 | 47.79 |
| Learned with random-image regularizer, $\lambda$=**4** | 56.89  (-2.25) | **47.95**  (+1.99) |
| Learned with random-image regularizer, $\lambda$=5 | 56.81 | 47.77 |
| Learned with random-image regularizer, $\lambda$=6 | 56.27 | 47.52 |
| Learned with random-image regularizer, $\lambda$=8 | 54.93 | 47.01 |
| Learned with random-image regularizer, $\lambda$=10 | 54.07 | 46.72 |
| Learned with random-image regularizer, $\lambda$=12 | 53.27 | 46.20 |

# C Distribution of answers per question type in VQA v2 and VQA-CP

Figure 6: Histograms of the ten most frequent answers for every question prefix in VQA v2 and VQA-CP. The last prefix is empty and is the "catch all" default. Stop words are omitted in the figure, making some prefixes appear identical, *e.g. What is* and *What is the*. Best viewed electronically with magnification.