[Reviews · NeurIPS 2020]

Review 1

Summary and Contributions: This paper provides an investigation of out-of-distribution generalization in visual question answering, as benchmarked by prior works on the VQA-CP dataset. The VQA-CP dataset by Agrawal et al. has different distributions in training and test, intentionally constructed so to encourage models to truly perform reasoning and generalize better, instead of naively picking up on question-only biases in the dataset. However, the authors demonstrate how several prior works on VQA-CP have (inadvertently) gamed this evaluation dataset without necessarily making progress due to a number of issues -- 1) exploiting knowledge of how the train/test splits were constructed to build models such that a) models are conditioned on the question prefix (and so will only work well on VQA-CP test and not generalize beyond), or b) poorly fit the training set. 2) the test set being used for model selection, and 3) either not reporting in-domain performance on VQA-CP train or reporting in-domain performance on VQA v2 val after retraining (which essentially makes it a separate model altogether). Next, the authors provide a few naive baselines that exploit the aforementioned issues (and as the authors acknowledge -- is not useful for any practical purposes) and perform well on VQA-CP test -- 1) a random predictions model that inverts the predicted answer distribution from training to test, and 2) a learned BUTD model that artificially ignores the top-predicted answer on VQA-CP test. The fact that a random predictions inverted model performs better on number and yes/no questions -- the question set that constitutes the largest fraction of performance -- is alarming, and provides a necessary and timely check on prior works on VQA-CP. Finally, the authors end with a few practical recommendations for future work on OOD generalization using VQA-CP -- 1) that future works should report in-domain performance as well on a held-out subset of VQA-CP train (so that higher performance on VQA-CP test is not because of fitting the training set poorly), 2) exclusive focus on "other" questions, where random prediction inverted baselines struggle, and 3) rigorously outline the source of performance improvements.

Strengths: This is an unconventional paper in that it does not present a novel task or algorithm (of practical significance), yet extremely important in that it provides a reality check for prior works on VQA-CP -- that they might be performing comparably to a randomly initialized model -- and so, several published papers might have been (inadvertently) flawed. The paper backs its findings by artificially constructing random baselines / regularizers that perform well on VQA-CP test, and goes on to provide concrete recommendations for future work in this space.

Weaknesses: I would encourage the authors to report some notion of statistical significance for results in Table 2 to get a better sense for what sort of differences are statistically meaningful. Other than that, the only weakness of this work is that it is a couple of years late. :) Out-of-distribution generalization is an important problem, and establishing accurate evaluation methodologies and rigorous baselines are drivers of scientific progress.

Correctness: To my understanding, the empirical methodology is consistent with the issues outlined, and experiments sufficiently validate the hypotheses.

Clarity: The paper is clearly written and provides a succinct set of baseline experiments to back its claims.

Relation to Prior Work: The paper is appropriately contextualized and provides a review of potentially problematic methodologies in prior work on VQA-CP.

Reproducibility: Yes

Additional Feedback:


Review 2

Summary and Contributions: This paper points out three issues of the VQA-CP dataset, a popular benchmark for the out-of-distribution testing in the domain of visual question answering: (1) relying on the known construction of the OOD splits; (2) using the test set for model selection; (3) evaluating in-domain performance after retraining. The authors also propose simple baselines that exploit these issues on VQA-CP and beat the state-of-the-art methods.

Strengths: 1. This paper targets on an important aspect of current machine learning methods, i.e., out-of-distribution testing, which deserves more research attention. 2. It is interesting and novel to see the issues on the VQA-CP datasets and it is important to share these insights with the community. 3. The three issues pointed out by this paper is reasonable and convincing. Besides, the simple baselines exploiting these issues, and the experiment results verify the hypotheses presented in this paper.

Weaknesses: The results in this paper focus on the VQA task and the VQA-CP dataset. I was wondering if other datasets for OOD evaluation have similar problems like VQA-CP. More discussion about other OOD datasets can further validate the hypotheses presented in this paper.

Correctness: Yes

Clarity: Yes

Relation to Prior Work: Yes

Reproducibility: Yes

Additional Feedback: I have read the responses from authors and the other reviews. My concern is well-addressed, so I recommend acceptance.


Review 3

Summary and Contributions: This paper first highlights 3 flaws in the existing VQA models and then proposes some simple yet efficient approaches that outperform the existing models on VQA-CP test dataset. Finally, suggestions for how to design benchmarks are provided.

Strengths: S1: the proposed flaws exist in the existing models and should receive more attention for the community of VQA. S2: The proposed approach is simple and efficient, which improves the performance on VQA-CP test from 38.82 to 55.37, achieving SOTA performance. S3: suggestions and discussions for using VQA-CP are given and pointing at future directions of VQA datasets

Weaknesses: W1: it seems that the proposed approach that employs random-image regularizer hurts the performs on the in-domain datasets, like VQA-CP validation and VQA v2 validation. W2: Do other VQA datasets have the same problem? Other datasets refer to GQA, VCR.

Correctness: Yes.

Clarity: Yes. I can easily follow the paper.

Relation to Prior Work: Yes.

Reproducibility: Yes

Additional Feedback: I have read the rebuttal and my questions are well answered, so I recommend accepting this paper.

[Author Response · NeurIPS 2020]

# Submission 180: Author Response

We thank the reviewers for their thoughtful comments. We are encouraged to receive positive feedback from all reviewers regarding the importance of investigating the value of out-of-distribution testing and discussion of the issues associated with it. Reviewers have described our work as "extremely important in that it provides a reality check for prior works on VQA-CP (R1)" and "targets on an important aspect of current machine learning methods (R2)". The reviewers also unanimously agreed on the clarity and correctness of the paper and our empirical evaluation. Reviewers have described our proposed baseline models and experimental validation as: "Interesting and novel" (R2), "simple and effective" (R3), "reasonable and convincing" (R2), and that the "experiments sufficiently validate the hypotheses"(R1).

The reviewers have also raised some concerns and issued some constructive suggestions about our work, which we address in the comments below. Reviewers' comments have been paraphrased for brevity.

***R3:*** *It looks like the random image regularizer hurts in-domain performance.*

**Response:** Firstly, the random image regularizer is designed to showcase how exploiting the fact that the test-set follows approximately the inverse distribution can be exploited and has no practical use case. As discussed in L258-260, it is possible to tune the trade-off between in-domain and out-of-domain performance by tuning the hyperparameter $\lambda$ for the random image regularizer. A part of the problem we want to highlight is that most algorithms do not even report the in-domain performance before retraining and our results clearly show that it is possible to obtain really high performance on the OOD split by sacrificing in-domain performance.

That being said, our random image regularizer works on-par or better than existing methods on in-domain performance reported before retraining [1, 2]. Table 4 only shows lambda = 5 and 12, which indeed show that it lags behind in in-domain setup (while far surpassing the OOD setup), but lower values of lambda (Figure 4a) shows that it is possible to have higher in-domain performance. E.g., at $\lambda = 2$, our results exceed [1] and are on par with [2] on both in-domain and out-of-domain splits.

***R3:*** *Do other VQA datasets (e.g., GQA, VCR) have the same problem?*

**Response:** Neither GQA nor VCR contain an OOD test split and therefore they are not capable of measuring OOD performance. However, if a split of the dataset was made in a manner similar to VQA-CP (lacking in-domain holdout set, OOD test set approximately inversely distributed as train) and the algorithms made similar assumptions (access to knowledge about the construction of test set, evaluating in-domain performance after retraining), our findings on the pitfalls of OOD testing would readily apply to any other VQA dataset. As discussed in our recommendation section, alleviating these concerns calls for a radically different approach to constructing datasets for OOD testing for VQA that is not present in any of the current VQA datasets to our knowledge.

***R2:*** *Do other datasets for OOD evaluation have similar problems like VQA-CP?*

**Response:** The pitfalls we identified with VQA-CP and the methods evaluated on it are relevant to OOD testing in general. The problems stem from the three key issues we discussed in the paper and are largely model and dataset agnostic. After our submission, some other papers [3] have pointed out similar issues with other datasets. We will summarize these observations in our final version.

***R1:*** *I encourage the authors to report some notion of statistical significance.*

**Response:** We agree with the spirit of the suggestion. In the final version, we will report statistical tests for variations of our baseline algorithms as well as comparison models, wherever appropriate.

# References

[1] Yonatan Belinkov, Adam Poliak, Stuart M Shieber, Benjamin Van Durme, and Alexander M Rush. Don't take the premise for granted: Mitigating artifacts in natural language inference. *arXiv preprint arXiv:1907.04380*, 2019.

[2] Anton van den Hengel Damien Teney, Ehsan Abbasnejad. Learning what makes a difference from counterfactual examples and gradient supervision. *arXiv preprint arXiv:2004.09034*, 2020.

[3] Ishaan Gulrajani and David Lopez-Paz. In search of lost domain generalization. *arXiv preprint arXiv:2007.01434*, 2020.


[Meta-Review · NeurIPS 2020]

All reviewers recommend the submission for acceptance and I agree. This work provided a detailed analysis of work on the VQA under Changing Priors problem -- pointing out a number of systematic deficiencies and providing a baseline that acutely demonstrates these shortcomings. The audience for this work is likely narrow but it will be important to that group.